# Effects of Voluntary Attention on Social and Non-Social Emotion Perception

**DOI:** 10.3390/bs13050392

**Published:** 2023-05-09

**Authors:** Hongtao Shao, Yang Li, Guiqin Ren

**Affiliations:** College of Psychology, Liaoning Normal University, Dalian 116029, China

**Keywords:** voluntary attention, social emotion, non-social emotion, perception

## Abstract

Existing studies have focused on the effect of emotion on attention, and the role of attention on emotion has largely been underestimated. To further determine the mechanisms underlying the role of attention on emotion, the present study explored the effects of voluntary attention on both social and non-social aspects of emotional perception. Participants were 25 college students who completed the Rapid Serial Visual Prime (RSVP) paradigm. In this study, the selection rates of participants’ emotional intensity, pleasure and distinctness perception of the pictures were measured. The results showed as following: (a) The cued condition selection rate was higher than the non-cued condition in the evaluation of non-social emotional intensity perception and pleasure perception, (b) In the evaluation of social emotional intensity and pleasure perception, there was no significant difference in the selection rate between the cued and non-cued condition, (c) The cued condition selection rate was higher than the non-cued condition in the perception of non-social positive emotional intensity and social negative emotional distinctness. The novel findings of this study revealed that the effect of voluntary attention on emotional perception is influenced not only by emotional valence but also by emotional sociality.

## 1. Introduction

There is a mutual effect between attention and emotional perception. Emotional information will trigger attention bias, and this attention bias has a possible role in maintaining and causally contributing to disordered affective states, such as anxiety and depression to a certain extent [1]. In the past, most studies mainly focused on the effect of emotional perception on attention, and found that emotions would affect the choice, direction, specificity, automation, control and resource consumption of attention [2,3,4,5,6]. Other researchers explored the role of emotion on attention in social attention mechanisms. For example, the observer’s recognition of target stimuli is affected by the direction of the gaze of others. The facial emotions of others have a moderating effect on the relationship between them [7]. However, not much attention has been paid to the effect of attention on emotional perception. It is expressed in two main aspects: on the one hand, attention can reduce the intensity of emotional perception, e.g., excessive exposure to negative emotions can trigger desensitization [8,9,10,11], which often occurs when participants passively attend to emotional stimuli; on the other hand, actively investing more attention towards resources will increase the intensity of emotional perception [12,13,14,15]. Compared with involuntary attention, voluntary attention is the process of active information selection. Exploring the effect of voluntary attention on emotion helps to explore the mechanism of emotional regulation and helps to provide a basis for clinical emotional intervention.

In past studies, researchers divided emotions into social emotions and non-social emotions based on sociality [16,17,18]. Social emotions are usually produced in social interactions between individuals and others; mainly referring to the emotions in social interactions, for example, people producing exciting emotions from seeing favorite stars. Non-social emotions are usually caused by incentives or disgusting stimuli with direct physiological correlations, which mainly reflect a basic biological driving force, such as people producing emotions from drinking sweet water. Social and non-social emotions are often induced by corresponding stimuli [19,20]. Weierich et al. noted that social-emotional stimuli contain person-relevant information, whereas non-social-emotional stimuli do not show person-relevant features, or person-relevant information appears as background information that is easily ignored [17]. Previous studies have shown that there are differences in the allocation of attention to social and non-social-emotional stimuli, with social-emotional stimuli preferentially capturing attention compared to nonsocial-emotional stimuli [21,22]. Social-emotional stimuli have more extreme pleasant or unpleasant ratings compared to nonsocial-emotional stimuli [23]. It has been demonstrated that increasing attentional investment in stimuli enhances the perceived emotional experience. Then, compared with non-social emotional stimuli, social emotional stimuli have a stronger emotional experience evaluation, whether or not it is caused by differences in attention distribution. Hence, this study is centered on the issue that was mentioned before, and thus aims to explore the effect of voluntary attention on social and non-social-emotional perception.

In addition, the different valences of social and non-social emotions will also affect the emotional experience [24,25,26]. Previous studies showed that different valences of emotions affect the process of attention [27]. Additionally, positive and negative emotions will attract more attention. Then, there is a question of whether the increasing attention input will also cause more extreme emotional experiences. To the benefit of deeply understanding the inherent mechanism of emotional processing by exploring this question, it can in turn provide new ideas for emotional regulation. Additionally, current studies on the relationship between attention and emotional perception have mainly focused on intensity perception, while this issue also deserves further in-depth exploration in the context of pleasure perception. Emotional perception includes different aspects, such as intensity perception and pleasure perception. Intensity represents the degree of emotional arousal, while pleasure represents the emotional valence. Those two are different dimensions of emotions [28,29,30]. Previous studies have found that voluntary attention significantly enhances emotional intensity perception [12], so whether it has the same role in pleasure perception is still unavailable to know.

Regarding the method of manipulation on voluntary attention, previous studies have found that the RSVP paradigm can spatially manipulate voluntary attention efficiently [31,32,33,34]. Therefore, this study adopts the RSVP paradigm, and aims to inspect voluntary attention with the effect of social and non-social-emotional perception through experiments. Additionally, attention can enhance stimulus representation based on the signal enhancement hypothesis [34,35,36]. Accordingly, this research assumes that voluntary attention significantly enhances the perception of negative, positive and neutral emotions. The focusing illusion theory states that when attention is focused on emotional perception, it is also affected by the context [37]. What is more, previous studies have pointed out that compared to non-social emotions, social emotions have a stronger emotional experience and attentional bias [38,39,40]. Therefore, another assumption of this study is that voluntary attention has different effects on social and non-social-emotional perception. Compared with non-social emotions, the experience of social-emotional perception is higher due to more attentional input.

## 2. Materials and Methods

### 2.1. Participants

This experiment adopted the G*Power 3.1 to estimate the sample capacity, and a 2 (cued: cued, non-cued) × 3 (emotion type: positive, negative, neutral) repeated measures ANOVA was used as a statistical test. According to the research of Faul et al. [41], these are the setting parameters: effect size *ƒ* = 0.25, α(Alpha, probability of Type I error) = 0.05, 1−β(Beta, probability of Type II error, 1−β is statistical power) = 0.80, and 19 people is the minimum sample capacity calculated by the number of measurements. Considering the possibility of invalid participants, 25 participants were recruited in this experiment. A total of 25 undergraduate and graduate students (11 males and 14 females) were randomly selected from Liaoning Normal University. All of them were right-handed with an average age of 26.10 ± 2.53 years old. All participants had normal or corrected-to-normal eyesight, no achromatopsia, no color blindness, and no history of neurological diseases or brain damage. Finally, we had twenty-three effective participants after excluding two participants whose answers did not meet the norms. The experiment was approved by the Ethics Committee of the School of Psychology, Liaoning Normal University. All participants signed a written consent form before the experiment was conducted.

### 2.2. The Experimental Design

A 2 (cued: cued, non-cued) × 3 (emotion type: positive, negative, neutral) × 2 (emotional sociality: non-social, social) three-factor within-subject experimental design was adopted for this experiment. To examine the effect of voluntary attention on the perception of non-social or socially different valence emotions, the experiment employed the RSVP paradigm to focus attention at a certain location.

### 2.3. Stimuli

The experimental materials came from the Complex Affective Scene Set (COMPASS) [8]. First, COMPASS clearly defines and classifies social and non-social attribute emotional images; secondly, COMPASS has wide applicability, because the participants are composed of diverse groups of people, including different races, different genders and different occupations; last, COMPASS balances not only the valence and the arousal of the image, but also controls the pixel size and contrast of the image.

#### 2.3.1. Non-Social Emotional Image Materials

In the experiment, three types of non-social emotional images (positive, negative, neutral) were first selected, and variance analysis was performed on their valence and arousal. In line with the hypothesis, there were significant differences in valence among the three types of images (*F*(2, 87) = 210.17, *p* < 0.001, *pη*^2^ = 0.83). After multiple comparisons, it was found that each of the two emotion types was significantly different (*p* < 0.001). In terms of arousal, there was a significant difference among the three types of images (*F*(2, 87) = 29.54, *p* < 0.001, *pη*^2^ = 0.40). However, the results showed that there was no significant difference between positive emotional images and negative ones after multiple comparisons (*F*(1, 58) = 0.01, *p* = 0.960). Additionally, a significant difference was found between positive and neutral images (*F*(1, 58) = 43.74, *p* < 0.001). Furthermore, a significant difference between negative emotional images and neutral ones was also found in this experiment (*F*(1, 58) = 43.11, *p* < 0.001).

According to the requirements of this experiment, the emotional images of non-social attributes were divided into two groups, and an independent sample t-test analysis was performed on the grouped images. Consistent with the hypothesis, no significant difference in valence between the two groups of negative emotional images was found (*t* = −0.56, *df* = 28, *p* = 0.96); and no significant difference in arousal was found (*t* = 0.14, *df* = 28, *p* = 0.470). Furthermore, no significant difference in valence between the two groups of positive emotional images was found (*t* =−0.93, *df* = 28, *p* = 0.964); and no significant difference in arousal was found (*t* = −0.40, *df* = 28, *p* = 0.759). Moreover, there was no significant difference in the valence of neutral emotional images between the two groups (*t* = 0.76, *df* = 28, *p* = 0.93); and no significant difference in arousal was found (*t* = 0.39, *df* = 28, *p* = 0.292).

#### 2.3.2. Social Emotional Image Materials

Secondly, three types of images (positive, negative, neutral) of social emotions were selected in our experiment, and ANOVA was performed on the valence and arousal. The results showed that there were also significant differences among three types of images in terms of valence (*F*(2, 87) = 180.67, *p* < 0.001, *pη*^2^ = 0.81). After multiple comparisons, it was found that each of the two emotional types was significantly different (*p* < 0.001). In terms of arousal, there were significant differences among the three types of images (*F*(2, 87) = 38.01, *p* < 0.001, *pη*^2^ = 0.47). There was no significant difference between positive and negative emotional images after multiple comparisons (*F*(1, 58) = 1.76, *p* = 0.45). However, a significant difference between positive and neutral images was found (*F*(1, 58) = 51.41, *p* < 0.001). There was also a significant difference between negative emotional images and neutral images (*F*(1, 58) = 72.63, *p* < 0.001).

The emotional images of each type of social attribute were divided into two groups according to the requirements, and an independent sample *t*-test analysis was performed on the grouped images. According to the results, there was no significant difference in valence between the two groups of negative emotional images (*t* = −0.30, *df* = 28, *p* = 0.873); and no significant difference in arousal (*t* = 0.60, *df* = 28, *p* = 0.367). There was also no significant difference in valence between the two groups of positive emotional images (*t* = −0.25, *df* = 28, *p* = 0.497); and no significant difference in arousal (*t* = −0.80, *df* = 28, *p* = 0.507). Additionally, no significant difference in the valence of neutral emotional images between the two groups was found (*t* = 1.32, *df* = 28, *p* = 0.424); and no significant difference in arousal (*t* = 1.31, *df* = 28, *p* = 0.956).

### 2.4. Experimental Process

E-prime 2.0 software was adopted to edit, run and collect data in this experimental program. There were 30 trials, and the basic flow of each trial was shown in Figure 1. Participants indicated their responses by pressing the up arrow (to select the image on the left) or the down arrow (to select the image on the right) on the keyboard to avoid spatial compatibility bias [42]. In addition, in order to ensure that the participants could recognize the cued and non-cued images, an image discrimination task was randomly arranged at the end of 20% of the trials. During each discrimination task, two images with the same valence and arousal as in the experiment were selected as disturbances.

Every trial consisted of the following steps (from top to bottom): (1) A 500 ms gaze point appears in the center of the screen. (2) A prompt appeared in the center of the screen, indicating that the next letter would appear on the left or right side. (3) A series of letters will appear at the top left or right of the screen. This sequence of letters presented 11 random letters one after the other, each letter presented for 200 ms. These constituted the RSVP paradigm. Participants were required to observe letters in RSVP and press the space key for the letter “X” that appeared. Among them, the probability of “X” was 20% [16,33]. (4) Two images of the same emotional type were presented on the left and right sides. The valence and arousal of the two images were effectively balanced. The display time of the two images for all was 2 s. The first 1 s contained hints (one of the images was located directly below the RSVP letter and presented at the same time as the letter), and the last 1 s did not. Studies have shown that through this process, the voluntary attention of the participants can be directed to the cue side [33]. (5) The participants needed to answer questions based on two images, such as “Which image seemed more emotionally intense?”. These included comparisons of the intensity, pleasure and distinctness of two images.

## 3. Results

The data were entered into SPSS22.0 statistical software for data analysis. In non-social and social emotion perception, a 2 (cue: cued, non-cued) × 3 (emotion type: positive, negative, neutral) repeated measure ANOVA were carried out on the selection rate of intensity perception, pleasure perception and distinctness perception, respectively. Simple effects analysis was performed when the interaction was significant, and a *p*-value of 0.05 was considered statistically significant. Partial eta squared (*pη*^2^) was calculated to estimate the size of the significant effects for ANOVAs.

### 3.1. The Effect of Voluntary Attention on Non-Social Emotional Perception

In the image discrimination task, the recognition accuracy rate of the cued images was about 93.33%, and the recognition accuracy rate of the non-cued images was about 95.83%, both of which were high. It indicated that participants were able to accurately identify both cued and non-cued images. A 2 (cue: cued, non-cued) × 3 (emotion type: positive, negative, neutral) repeated-measures ANOVA was performed on the selection rates of intensity perception, pleasure perception and distinctness perception. The experimental results of the effect of voluntary attention on non-social emotional perception were shown in Table 1. Statistics results of Voluntary Attention on the Emotional Perceived Enhancement Effect of Non-Socially Different Valences were shown in Table 2. It was found that in the evaluation of emotional intensity perception, the main effect of the cued images was significant (*F*(1, 19) = 4.58, *p* = 0.046, *pη*^2^ = 0.19); that is, the selection rate of the emotional intensity perception of cued images (56.3%) was significantly higher than that of non-cued images (43.7%); there was a significant interaction of the cued and emotional type (*F*(2, 38) = 4.58, *p* = 0.025, *pη*^2^ = 0.18). Simple effect analysis found that in the perception of positive emotional intensity, the selection rate of cued images (64.5%) was significantly higher than that of non-cued images (35.5%) (*p* = 0.003, *pη*^2^ = 0.39); in terms of the intensity perception of negative emotional images, there was no significant difference in the selection rate (*p* = 0.635, *pη*^2^ = 0.01); in terms of the intensity perception of neutral emotional images, there was no significant difference in the selection rate of cued images and non-cued images (*p* = 0.543, *pη*^2^ = 0.02). In addition, it was suggested that the emotional intensity perception of the image was tested on the emotional type, and it was found that the main effect of the emotional type was significant (*F*(2, 38) = 4.05, *p* = 0.025, *pη*^2^ = 0.18); further analysis found that positive emotions and negative emotions were not significantly different (*p* = 0.102, *pη*^2^ = 0.25); no significant difference between positive and neutral emotions (*p* = 0.089, *pη*^2^ = 0.25); no significant difference between negative and neutral emotions (*p* = 0.908, *pη*^2^ = 0.25).

In the evaluation of emotional pleasure perception, the main effect of the cued images was significant (*F*(1, 19) = 10.29, *p* = 0.005, *pη*^2^ = 0.35); that is, the emotional pleasure perception selection rate (55.5%) of the cued images was significantly higher than that of the non-cued images (44.5%); the interaction between the cued images and emotional type was not significant (*F*(2, 38) = 1.58, *p* = 0.220, *pη*^2^ = 0.08); there was no significant difference in the emotional type on the pleasure perception of the cued images (*p* = 0.220, *pη*^2^ = 0.08). In the evaluation of image distinctness perception, the main effect of the cued images was not significant (*F*(1, 19) = 0.09, *p* = 0.772, *pη*^2^ = 0.01); the interaction between cued and emotion type was not significant (*F*(2, 38) = 1.91, *p* = 0.163, *pη*^2^ = 0.09); there was no significant difference in the image distinctness perception in the emotion type (*p* = 0.163, *pη*^2^ = 0.09).

### 3.2. The Effect of Voluntary Attention on Social-Emotional Perception

In the image discrimination task, the accuracy of cued images recognition was about 95%, and the accuracy of non-cued images recognition was about 85.83%, both of which were high, which indicated that the participants were able to recognize both cued and non-cued images accurately. A 2 (cue: yes, no) × 3 (emotion type: positive, negative, neutral) repeated measures ANOVA was conducted for the selection rates of intensity perception, pleasure perception and distinctness perception, respectively. The experimental results of the effect of voluntary attention on non-social emo-tional perception were shown in Table 3. Statistics results of Voluntary Attention on the Emotional Perceived Enhancement Effect of Socially Different Valences were shown in Table 4. The results found that the main effect of the cued images was not significant on the emotional intensity perception evaluation (*F*(1, 19) = 2.50, *p* = 0.130, *pη*^2^ = 0.12); the interaction between the cued and emotional type was not significant (*F*(2, 38) = 0.528, *p* = 0.594, *pη*^2^ = 0.03). In addition, the emotional intensity perception of cued images was tested on emotional type, and the results found that the main effect of emotional type was not significant (*F*(2, 38) = 0.53, *p* = 0.594, *pη*^2^ = 0.03).

On the perceived evaluation of emotional pleasure, the main effect of the cued images was not significant (*F*(1, 19) = 1.29, *p* = 0.270, *pη*^2^ = 0.06); the interaction between the cued images and emotional type was not significant (*F*(2, 38) = 1.72, *p* = 0.192, *pη*^2^ = 0.08); there was no significant difference in the emotional type on the pleasure perception of the cued images (*F*(2, 38) = 1.72, *p* = 0.192, *pη*^2^ = 0.08). The main effect of the cued images was not statistically significant in the perceived evaluation of image distinctness (*F*(1, 19) = 3.35, *p* = 0.051, *pη*^2^ = 0.19), but the selection rate of perceived distinctness of cued images (53.7%) showed a slightly higher trend than that of non-cued images (46.3%); the interaction between the cued images and emotional type was significant (*F*(2, 38) = 4.28, *p* = 0.021, *pη*^2^ = 0.18). Simple effects analysis revealed that the selection rate on the negative emotion image distinctness perception was significantly higher on the cued side (60.5%) than on the non-cued side (39.5%) (*p* = 0.002, *pη*^2^ = 0.42). In the results of the cued image distinctness perception, it was found that the negative emotional selection rate (60.5%) was higher than that of the neutral images (47.5%) (*p* = 0.011, *pη2* = 0.37).

## 4. Discussion

Our study explored the effects of voluntary attention on the perception of social and nonsocial emotions by using the RVSP paradigm. The results demonstrated that voluntary attention significantly enhanced the perception of intensity and pleasure of non-social emotions, but not the perception of distinctness. In contrast, the enhancement of social emotional perception by voluntary attention was not significant. The results confirmed the hypothesis of our study that there are differences in the effects of voluntary attention on the perception of social and nonsocial emotions, indicating that the enhancement effect of voluntary attention on emotional perception is influenced by emotional sociality. The results are consistent with previous research in that voluntary attention enhances the representation of stimuli [43,44]. For example, Mrkva et al. examined the effect of attention on emotion perception by manipulating cueing cues [12]. It was found that individuals perceived the emotional intensity of cued stimuli more strongly compared to non-cued stimuli. This suggests that voluntary attention has a significant effect on the enhancement of emotional intensity perception. Inconsistent with the results of previous studies, this study found a significant enhancement of voluntary attention on pleasure perception. In contrast, Mrkva et al. found that there was a significant enhancement of individual voluntary attention on perception of emotional intensity and perception of distinctness, but no significant effect on pleasure perception [12]. The difference is most likely related to the characteristics of the experimental materials. In this study, the experimental material was divided into social and non-social emotional images, whereas in the study by Mrkva et al. [12], they did not take the emotional sociality into account for the experimental material. For instance, it shows that it is necessary to make a distinction between social and non-social emotional images.

The results of this study suggested that voluntary attention had a significant effect on the enhancement of perceived intensity and perceived pleasure of non-social emotions, but not on the perception of these two aspects of social emotions. There are two possible reasons. One is that there is inconsistency between social and nonsocial emotions in the subjective experience of individuals [45,46], as Britton et al. noted that individuals perceive social-emotional stimuli more strongly than non-social-emotional stimuli of equal arousal [46]. In this study, when participants performed contrastive perception of social-emotional stimuli, it was likely that they perceived both contrasting stimuli strongly, which resulted in a less significant enhancement of voluntary attention; second, the less significant enhancement of voluntary attention on the perception of social-emotional intensity and pleasure perception may be due to insufficient attentional input resources. For example, Mrkva et al. examined the effect of attention on stimulus selection priority by manipulating the frequency of attention. It was found that individuals tended to prioritize stimuli that received three times focuses of attention compared to stimuli that received only one time focus of attention. It suggests that an individual’s preference for a stimulus is influenced by attentional resources, and the more attention is devoted to a stimulus, the more stimuli will be preferred [47]. Thus, emotional perception may be influenced by attentional resources, and increasing attentional input may elicit stronger emotional perception.

In addition, the results illustrated that there is a link between intensity perception and pleasure perception. Intensity corresponds to arousal and pleasure corresponds to validity. Some studies have argued that arousal and valence exist independently. For example, Vogt et al. pointed out that arousal affects the attentional allocation of emotional images and is not affected by validity [48]. In contrast, Bradley et al. proposed the motivational model of affect, arguing that valence and arousal are related to some extent and are not independent dimensions [49]. Our study supports the motivational model of affect in that arousal and emotion exist correlated. Furthermore, Keltner and Kring proposed a social function of emotion [50], a view that emphasizes the highly dependent link between social and emotion, a link that suggests that social emotions have a more distinct social meaning for individuals relative to non-social emotions. The signal enhancement hypothesis suggests that attention enhances stimulus representations [36]. Perceived amplification of the distinctiveness of social-emotional images is enhanced under voluntary attention conditions. It provides evidential support for this study, suggesting a tendency for intentional attention to enhance the distinctness perception of social images, while the enhancement of the distinctness perception of non-social images was not significant.

This study also found a significant effect of voluntary attention on the enhancement of perceived intensity and perceived pleasure of non-social positive emotions. This is consistent with the hypothesis and previous research that voluntary attention has a significant effect on the enhancement of emotion perception [12]. Furthermore, this study did not find a significant enhancement effect of voluntary attention on the perception of neutral and negative emotions, which may be due to the reverse effect of emotions. Previous studies have pointed out that attentional manipulation is influenced by emotional types [31]. This is reflected in the presence of attentional bias for negative emotions compared to positive emotional perceptions. Therefore, the manipulation of attention will be more difficult in contrast to negative emotional images. However, neutral emotions are less attractive to attention and more susceptible to other stimuli. When comparing negative and neutral emotional images, there would be interference from the non-cued images to the cued images, so that it would affect the participants’ attentional engagement to the cued images. It ultimately resulted in a nonsignificant effect of voluntary attention on the perceived enhancement of negative and neutral emotions. Voluntary attention had a more significant enhancement effect on the perception of the distinctness of social negative emotional images compared to socially neutral emotions. In contrast, neutral emotional images lacked more distinctness for both positive and negative emotional images [51]. Focused illusion theory suggests that the perception of attended stimuli is enhanced relative to non-attended stimuli [52,53,54,55], and the distinctness of negative emotional images is inherently greater than neutral emotional images. It is not difficult to conclude that voluntary attention has a more significant enhancing effect on the distinctness perception of negative social-emotional images.

There are some shortcomings in this study, and these issues need to be further addressed in future research. First of all, the technical means of this research need to be further improved, and the latest technology can be applied to carry out research in this field in the future. This study was based on the subjective experience of the participants. In the future, the brain-computer interface can be used to further verify the objective indicators, such as EEG, skin electricity and heart rate that can measure the emotional experience of the participants. Through the combination of virtual reality (VR) and eye movement, the attention allocation status of the participants can be examined with more ecological validity and intuition. Second, this study used a like-emotion comparison. In contrast, in real situations, individuals are more often confronted with different emotional types. Future studies should focus on examining the correlation between attention and emotion under conditions of different emotional contrast. In addition, the sample size of this study was relatively small, so future research can appropriately expand the sample size. Moreover, it is worth exploring whether there is an enhancement effect of voluntary attention on different sensory channels. Previous studies have found that emotional recognition is influenced by different channels [56,57,58]. Therefore, future research could explore whether the role of attention on emotional perception is influenced by different channels.

The enhancement effect of attention on emotions will be affected by the social nature of emotions, so whether the causes and treatment of patients with depression or anxiety can be considered from this research perspective is something that can be verified in the future. Some researchers have found that emotional perception is affected by others’ gaze or facial direction, such as the gaze-liking effect; that is, when people see others looking at an object, they tend to evaluate it more positively and express a higher degree of liking for it [59]. Whether or not the mechanism behind this is due to attention-enhancing emotional pleasure is something that can be examined in future research. Other fields and research methods of emotional perception can also be combined with the focus of this study. For example, previous research has developed emotional face recognition technology [60]. In the future, we can also improve relevant emotional recognition technology by establishing neural models of the influence of attention on emotions and combining computer modeling.

## 5. Conclusions

Through the RSVP paradigm, we explored the effects of voluntary attention on emotional perception to further determine the mechanism of emotional regulation and to provide a basis for clinical emotional intervention. Different from previous researchers investigating the process of how voluntary attention drives emotion, we focused on how the processing of voluntary attention drives social and non-social emotion. The results demonstrated that voluntary attention significantly enhanced the perception of intensity and pleasure of non-social emotions, but not the perception of distinctness. In contrast, the enhancement of social emotional perception by voluntary attention was not significant. Notably, voluntary attention significantly enhanced the perception of non-social positive emotional intensity and social negative emotional distinctness. Our study revealed how the correlation between college students’ voluntary attention on emotional perception is not only affected by emotional valence, but also by emotional sociality.

## Figures and Tables

**Figure 1 behavsci-13-00392-f001:**
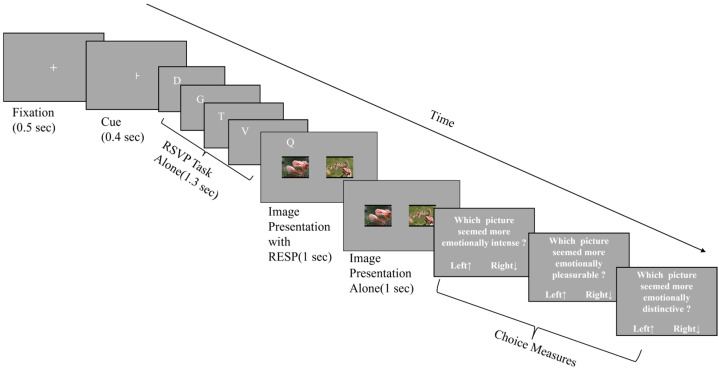
Graphical representation of the experimental design.

**Table 1 behavsci-13-00392-t001:** Descriptive Statistics of Voluntary Attention on the Enhancement Effect of Non-social Emotional Perception.

	Cued M (SD)	Non-Cued M (SD)	Cued Main Effect	Cue × Emotion Type Interaction
			F (*p*)	F *(p)*
**Intensity perception**	0.56 (0.03)	0.44 (0.03)	4.58 (0.046)	4.05 (0.025)
**Pleasure perception**	0.56 (0.02)	0.45 (0.02)	10.29 (0.005)	1.58 (0.220)
**Distinctiveness perception**	0.51 (0.03)	0.49 (0.03)	0.09 (0.772)	1.91 (0.163)

Note: M = Mean, SD = Standard Deviation, F = Fisher-Snedecor distribution, *p* = *p*-value.

**Table 2 behavsci-13-00392-t002:** Descriptive Statistics of Voluntary Attention on the Emotional Perceived Enhancement Effect of Non-Socially Different Valences.

	Cued Positive Emotion M (SD)	Cued Negative Emotion M (SD)	Cued Neutral Emotion M (SD)	Non-Cued Positive Emotion M (SD)	Non-Cued Negative Emotion M (SD)	Non-Cued Neutral Emotion M (SD)
**Intensity perception**	0.65 (0.04)	0.52 (0.04)	0.53 (0.04)	0.36 (0.04)	0.48 (0.04)	0.48 (0.04)
**Pleasure perception**	0.61 (0.04)	0.54 (0.04)	0.52 (0.02)	0.39 (0.04)	0.47 (0.04)	0.48 (0.02)
**Distinctiveness perception**	0.56 (0.04)	0.50 (0.04)	0.47 (0.04)	0.44 (0.04)	0.50 (0.04)	0.54 (0.04)

Note: M = Mean, SD = Standard Deviation.

**Table 3 behavsci-13-00392-t003:** Descriptive Statistics of Voluntary attention on the enhancement effect of so-cial-emotional perception.

	Cued M (SD)	Non-Cued M (SD)	Cued Main Effect	Cue × Emotion Type Interaction
			F (*p*)	F (*p*)
**Intensity perception**	0.47 (0.02)	0.53 (0.02)	2.502 (0.130)	0.528 (0.594)
**Pleasure perception**	0.52 (0.02)	0.48 (0.02)	1.292 (0.270)	1.723 (0.192)
**Distinctiveness** **perception**	0.54 (0.02)	0.46 (0.02)	4.346 (0.051)	4.279 (0.021)

Note: M = Mean, SD = Standard Deviation, F = Fisher-Snedecor distribution, *p* = *p*-value.

**Table 4 behavsci-13-00392-t004:** Descriptive statistics of voluntary attention on the emotional perceived enhancement effect of socially different valences.

	Cued Positive Emotion M (SD)	Cued Negative Emotion M (SD)	Cued Neutral Emotion M (SD)	Non-Cued Positive Emotion M (SD)	Non-Cued Negative Emotion M (SD)	Non-Cued Neutral Emotion M (SD)
**Intensity** **Perception**	0.45 (0.03)	0.48 (0.03)	0.50 (0.04)	0.56 (0.03)	0.52 (0.03)	0.51 (0.04)
**Pleasure** **Perception**	0.52 (0.03)	0.56 (0.03)	0.49 (0.02)	0.49 (0.03)	0.45 (0.03)	0.52 (0.02)
**Distinctiveness** **Perception**	0.53 (0.04)	0.61 (0.03)	0.48 (0.03)	0.47 (0.04)	0.40 (0.03)	0.53 (0.03)

Note: M = Mean, SD = Standard Deviation, F = Fisher-Snedecor distribution, *p* = *p*-value.

## Data Availability

The data and codes of this study can be found at the following address: https://www.scidb.cn/surl/xlxb (accessed on 1 May 2022).

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
