# Peer review of "Effects of Voluntary Attention on Social and Non-Social Emotion Perception"

_behavsci, 2023, doi:10.3390/bs13050392_

Round 1

Reviewer 1 Report

1) Figure 1: Is labelled in Chinese please change to English or use dual language.

2) After line 161 please rephrase the sentence.

3) All symbols used show be abbreviated and explained in a table. (Ex: t, df, p, F )

4) Short come of this study can be tabulated and directed to scope for future work sub section.  

5) Conclusion can be improved.          

6) Existing work on in the same direction but with different perceptive like the following  can be used to improve the reader refences.

7) Section 2, need to be change and also please highlight the experiment as equations.

8) You can also so include past study on emotional perception in other domain and methods like

https://doi.org/10.1016/j.jksuci.2018.09.011

Reviewer 2 Report

1. The references are too old and lack an introduction to the latest technology in the field.

2. Whether there is too little data on 23 individuals, and whether it can reveal the laws studied in the article from a statistical perspective.

Reviewer 3 Report

The paper evaluates the interaction between attention, emotional cues, positive emotions, negative emotions, non-social and social emotions. Unlike previous studies, the research focused on how voluntary attention affects social and non-social emotions. The paper is well organized however the Figure 1 (section 2.4 experimental process section) caption is excessively large. The reviewer recommends moving it into its own paragraph and keeping the caption concise. The symbols used in 2.3.1 and 2.3.2 should be described properly. For example, F, p, df, alpha, beta etc should be defined wherever they are referenced for the first time. Overall, the findings are interesting, however the number of participants was too small and should be identified as a limitation in the future scope section.

Reviewer 4 Report

This is a study exploring the role of attention in emotion. A sample (N=25) completed and RSVP task. The main results showed that a link between attention and emotion can actually be detected.

I only have a few minor comments.

1) Line 27: The role of emotion on attention has been largely explored also in social attention mechanisms (Dalmaso et al., 2020) , and this should be briefly mentioned; in addition, I would suggest also add a missing, relevant, review focused on emotion and attention (Yiend, 2010).

2) Results sections: I suggest to always add the effect sizes after ANOVA (partial eta squared) and t-tests (Cohen d).

3) From line 317: I would also add the idea to use eye movements (other than EEG or psychophysiology measures) to explore this topic, as they can provide a more direct and ecological measure of attention allocation.

References

Dalmaso, M., Castelli, L., & Galfano, G. (2020). Social modulators of gaze-mediated orienting of attention: A review. Psychonomic Bulletin & Review, 27(5), 833–855. https://doi.org/10.3758/s13423-020-01730-x

Yiend, J. (2010). The effects of emotion on attention: A review of attentional processing of emotional information. Cognition & Emotion, 24(1), 3–47. https://doi.org/10.1080/02699930903205698

Round 2

Reviewer 1 Report

Figure 1 : is too dark , please inform for better readability. 

Author Response

Thanks for your suggestion, we have modified Figure 1 and changed the font color in the picture to white.

Reviewer 4 Report

I am happy with this new version.

Author Response

Thanks for your valuable advice, kind regards